# Effect of *Staphylococcus aureus* Contamination on the Microbial Diversity and Metabolites in Wholewheat Sourdough

**DOI:** 10.3390/foods11131960

**Published:** 2022-07-01

**Authors:** Weidan Guo, Zhengwen Li, Xiangjin Fu, Wenhua Zhou, Jiali Ren, Yue Wu

**Affiliations:** 1College of Food Science and Engineering, Central South University of Forestry and Technology, Changsha 410004, China; guoweidan96@163.com (W.G.); lzw970214@163.com (Z.L.); zhowenhua@126.com (W.Z.); jialiren_zhen@hotmail.com (J.R.); wuyuejn@163.com (Y.W.); 2College of Food Science and Engineering, South China University of Technology, Guangzhou 510000, China; 3Hunan Key Laboratory of Processed Food for Special Medical Purpose, Changsha 410004, China; 4Hunan Key Laboratory of Forest Food Processing and Safety Quality Control, Changsha 410004, China; 5Hunan Engineering Technology Research Center of Nutrition and Health Products, Changsha 410004, China

**Keywords:** *Staphylococcus aureus*, sourdough, γ-aminobutyric acid, metabolites

## Abstract

Wholewheat sourdough products are becoming increasingly more popular, and *Staphylococcus aureus* is a common opportunistic pathogen in dough products. The effects of *S. aureus* contamination (10^2^ cfu/g) on metabolites as well as titratable acidity (TTA), pH, and microbial diversity of sourdough were investigated. *S. aureus* contamination significantly decreased the content of mannose while increasing the sorbitol in sourdough (*p* < 0.05). The *S. aureus* contamination significantly reduced the number of lactic acid bacteria (LAB), such as *Lactobacillus curvatus*, and the TTA values (*p* < 0.05). Furthermore, *S. aureus* contamination significantly reduced the content of most esters and acid flavor compounds while significantly increasing the content of 2,4-decadienal (*p* < 0.05), which is a compound that could have a negative impact on the flavor of sourdough. The PCA model developed based on volatile metabolites data could be used to distinguish contamination of *S. aureus* in sourdough cultured for 4 h. Sorbitol, 2,3-dimethylundecane, 1-pentanol, and 3-methylbutanoic acid were newly found to be the characteristic metabolites in *S. aureus*-contaminated sourdough.

## 1. Introduction

Sourdough products have special flavor and health benefits generated during fermentation by yeast and lactic acid bacteria (LAB) [1,2]. For instance, the intake of sourdough fermented foods replenishes the number and diversity of intestinal microbial flora, which may benefit health [3]. On the other hand, wholewheat products contain a large amount of vitamins and dietary fiber, which have health benefits as well [4]. Therefore, wholewheat sourdough products combine the health benefits of wholewheat and sourdough, receiving increasing attention [5]. Ma, et al. [3] suggested that habitual consumption of wholewheat sourdough bread helps reduce the risk of coronary heart disease, diabetes, and cancer.

The process of spontaneous sourdough is an open system; thus, it could be easily contaminated by bacteria, such as *Staphylococcus aureus*, which is a common opportunistic pathogen in dough products [6]. *S. aureus* is a commensal human pathogen colonizing the nose and skin of flour handlers, who are considered the potential asymptomatic carriers or sources of dough contamination [7]. The number of *S. aureus* cells is one of the necessary sanitary inspection indexes of fresh dough products (GB19295-2011) [8]. It has also been shown to grow in pork, milk rice cakes, etc. [9,10,11].

Microbial contamination has a significant effect on food compositions, which were too complex to be comprehensively clarified [5]. Metabolomics can comprehensively analyze the composition of small molecules in foods, thus becoming a new method for evaluating food quality and mining bio-marks of microbial contamination [12,13]. In this study, the GC-MS-based metabolomic approaches were used to investigate the effect of *S. aureus* contamination on the metabolites of sourdough in an effort to find the biomarkers for S. *aureus* contamination, which might be helpful for the development of a rapid S. *aureus* detection method.

There are many types of *S. aureus*, and according to national food safety inspection standard (GB 4789.10-2016, China) [14] and literatures [9], *S. aureus* CICC21600 was often used as a positive control strain in food contamination studies and quality control studies. The optimal growth temperature of S. *aureus* CICC21600 is 37 °C. It is facultative anaerobe and produces enterotoxin [15]. Thus, *S. aureus* CICC21600 was used as the contaminating strain in this study.

## 2. Materials and Methods

### 2.1. Culture of S. aureus

*Staphylococcus aureus* CICC21600 (China Center of Industrial Culture Collection, Beijing, China) was inoculated in trypticase soy broth (TSB; Qing Dao Hope Bio-Technology Co. Ltd., Qingdao, China) and grown at 37 °C overnight with agitation of 200 r/min. The cell count was determined using 10-fold dilution plating method according to the standard of GB4789.10-2016 (China), by which *S. aureus* was cultured on Baird–Parker plate (Beijing Land Bridge Technology Co. Ltd., Beijing, China) at 37 °C for 48 h.

### 2.2. Preparation of Sourdough and Contamination with S. aureus

The sourdough was prepared in two stages, including seed dough preparation and sourdough preparation, according to a traditional method [16,17] with modifications. 

Flour (Xinxiangliangrun Wholegrain Food Co. Ltd., Xinxiang, China) was mixed with water at a 1:1 ratio (w/w), noted as fresh dough, and then incubated at 25 °C for 24 h, noted as the 1st-seed sourdough, which was added into another fresh dough at 20% (w/w) and then incubated at 25 °C for 24 h to obtain the 2nd-seed dough, and the process was repeated for one more time to obtain the 3rd-seed dough, which was then freeze-dried. The *S. aureus* count in the dried 3rd-seed dough was analyzed (GB4789.10-2016, China) [14]. The dried 3rd-seed dough that did not contain *S. aureus* was used to make sourdough.

The flour, water, and dried 3rd-seed sourdough were mixed at a 60:35:5 ratio. The *S. aureus* culture (10^4^ cfu/mL) was added into the dough at 1% (v/w) for spiking samples, while the same volume of sterile water was added for non-contaminated control samples. The dough was subsequently incubated at 25 °C for 16 h to obtain sourdough samples.

### 2.3. Growth of S. aureus in Sourdough

Fermented sourdough was sampled at 0, 2, 4, 8, 12, and 16 h. The colony formation unit (cfu/g) of *S. aureus* in the samples was determined by 10-fold dilution plating method according to GB4789.10-2016 (China) [14].

### 2.4. Determination of Total Titratable Acidity (TTA) and pH of Sourdough

Ten grams of dough were homogenized with 90 mL of distilled water for 3 min; then, the total titratable acidity (TTA) was measured and expressed as the amount (mL) of 0.1 N NaOH to achieve a pH of 8.5. The pH values were determined by a pH-meter (PHS-3E) (INESA Scientific Instrument Co., Ltd., Shanghai, China) [18].

### 2.5. Analysis of Bacterial Composition

Total bacterial DNA in sourdough was extracted by Omega bacterial genomic DNA extraction kit. PCR amplification (ABI GeneAmp^®^ 9700) and high-throughput DNA sequencing (Miseq platform, Illumina, San Diego, CA, USA) were entrusted to Shanghai Majorbio Technology Co., Ltd. (Shanghai, China), using primers 338F (5′-ACTCCTACGGGAGGCAGCAG-3′) and 806R (5′-GGACTACHVGGGTWTCTAAT-3′) [19].

### 2.6. Analysis of Metabolites

The sample (50 ±1 mg) was mixed with 450 μL of pre-cooled methanol: dH_2_O (3:1, v:v) and 5 μL of internal standard (L-2-chlorophenylalanine, 1 mg/mL stock). The mixture was vortexed for 30 s, homogenized using a ball mill at 35 Hz for 4 min, and then ultrasonicated in ice water bath for 5 min. After centrifugation at 8000× *g* at 4 °C for 15 min, 150 μL of the supernatant was transferred to a fresh tube, evaporated in a vacuum concentrator, added with 100 μL of methoxyamination hydrochloride (20 mg/mL in pyridine), incubated at 80 °C for 30 min, and then derivatized with 120 μL of bis(trimethylsilyl)trifluoroacetamide (containing 1% Trimethylchlorosilane, *v/v*) at 70 °C for 1.5 h [20].

All samples were subsequently analyzed by a gas chromatograph coupled with a time-of-flight mass spectrometer (GC-TOF-MS, Agilent 7890) using a Rtx-5ms capillary column (30 m × 250 μm × 0.25 μm; SHIMADZU, Kyoto, Japan). The sample (1 μL) was injected in splitless mode. Helium was used as the carrier gas, the front inlet purge flow rate was 3 mL/min, and the gas flow rate through the column was 1 mL/min. The initial temperature was kept at 50 °C for 1 min, then raised at 10 °C/min to 310 °C, and held for 8 min. The injection, transfer line, and ion source temperatures were 280, 280, and 250 °C, respectively. The ionization was in electron impact mode at an energy of −70 eV. The mass spectrometric data were acquired in full-scan mode at an *m/z* range of 50–500 amu at a rate of 12.5 spectra per second after a solvent delay of 6.33 min.

### 2.7. Analysis of Volatile Metabolites (VOMs) in Sourdough

VOMs were analyzed using a head space SPME-GC-MS: 0.5 g of sourdough was mixed with 0.5 mL of water, 0.5 g of NaCl, and 10 μL of 2-octanol (100 μg/mL, internal standard) in a 10 mL sample bottle (Supelco, Bellefonte, PA, USA). The mixture was magnetic stirred (750 r/min) at 50 °C for 10 min and then subjected to solid-phase microextraction (SPME) using the DVB/Carboxen/PDMS (50/30 μm; Supelco, Bellefonte, PA, USA) fiber with the exposure time of 30 min.

GC-MS was conducted using GCMS-QP2010 Plus (SHIMADZU, Kyoto, Japan) equipped with a capillary column (Rtx-5ms, 30 m × 0.25 mm × 0.25 μm; SHIMADZU, Kyoto, Japan). The operating parameters were set as follows: splitless mode; the injection, transfer line, and ion source temperatures were 230, 280, and 230 °C, respectively; carrier gas (helium) flow at 1.0 mL/min; and EI, 70 eV. The oven was programmed as follows: 35 °C held for 2 min, increased to 180 °C at 5 °C/min, and then increased to 250 °C at 20 °C/min and held for 5 min. The MS scan range was 30–1000 amu [20].

### 2.8. Data Analysis

Analysis of GC-MS raw data, including peak extraction, baseline adjustment, deconvolution, alignment, and integration, was carried out on Chroma TOF (V 4.3x, LECO) software. Metabolites identification was conducted by matching the mass spectrum and retention index of the samples with those of the standards available in the LECO-Fiehn Rtx5 database and NIST05 libraries.

All experiments were run in sextuplicate (*n* = 6). The ANOVA (one-way ANOVA) was conducted using SPSS 22.0 (IBM, Armonk, NY, USA). Principal component analysis (PCA) was carried out using SIMCA-13.0 software (Umetrics, Aubagne, France) to analyze the data of sourdough VOMs. The heatmap was created using “heatmap2” package in R. Bacterial composition analysis was performed using Majorbio Cloud Platform (www.majorbio.com, accessed on 14 June 2021), a free online platform.

## 3. Results and Discussion

### 3.1. The Composition of LAB in Sourdough

Main bacteria in sourdough that were identified included *Lactobacillus curvatus*, *Pediococcus pentosaceus*, and *Leuconostoc citreum* (Figure 1), all of which were reported have health benefits [21]. The cell count of *L. curvatus* in contaminated sourdough decreased, while that of *P. pentosaceus* (*p* < 0.05) and *L. citreum* notably increased. The composition of microflora in sourdough can vary with dough composition, fermentation temperature and time, process conditions, etc. [1,22]. *L**. curvatus* has also been found to be the main LAB in Italian and Brazilian sourdoughs [22,23].

### 3.2. Growth of S. aureus in Sourdough

The growth curve (Figure 2) showed that the number of *S. aureus* cells was 10^3^ cfu/g after about 3 h of culturing, which was the limit set by China’s national standard (GB19295-2011, China), and was about 10^6^ cfu/g after 8 h of culturing and was kept constant thereafter, indicating that the cell growth was inhibited in this period.

Many reports on the growth characteristics of *S. aureus* in different food matrixes have been published. The level of *S. aureus* in no-knead bread dough fermented by yeast has been shown to increase by 0.4, 1.1, 1.7, and 2.2 log cfu/g after 18 h culturing at 21 °C, 27 °C, 32 °C, and 38 °C, respectively [24]. The growth of *S. aureus* in rice cake stored at 25 °C and 35 °C reached the stationary phase within 24 and 16 h, respectively [9], whereas that in pork stored at 37 °C reached the stationary phase in about 14 h [10]. Additionally, the growth of *S. aureus* in milk stored at 25 °C reached the stationary phase in 10 h [11] In this study, the growth of *S. aureus* in sourdough (25 °C) reached the stationary phase in 8 h (Figure 2), indicating that *S. aureus* could grow rapidly at the initial stage in wholewheat sourdough compared to rice cake, pork, and milk. This might be because LAB, including *L. curvatus* and *Ln. citreum*, which are the heterofermentative species, in sourdough could effectively convert fructose into mannitol [25], which is conducive to the growth and proliferation of *S. aureus*.

The highest *S. aureus* cell number in sourdough was only about 10^6^ cfu/g, which is 10 to 100 times lower than the numbers of *S. aureus* in pork and milk, respectively [10,11] This indicates that the growth *S. aureus* in sourdough was inhibited after 8 h [26]. One characteristic of LAB is the ability to produce acid. This is beneficial to decrease *S. aureus* risk, as the growth of *S. aureus* is inhibited by low pH values. Furthermore, bacteriocins secreted by LAB (such as *P. pentosaceus*) can also inhibit the growth of *S. aureus* [27]. However, because the number of *S. aureus* cells exceeding 10^5^ cfu/g can lead to production of enterotoxins that can cause human diseases [11], the contamination by *S. aureus* (even at a number as low as 10^2^ cfu/g) in sourdough could thus pose risks.

### 3.3. The Effect of S. aureus Contamination on pH and TTA

The pH of sourdough decreased to 4.0–4.1 within 16 h of fermentation (Figure 3), which is consistent with the observation reported in the literature [28]. Additionally, the pH values of the two groups were not significantly different (*p* > 0.05).

The TTA of sourdough increased rapidly, from 1.40 ± 0.13 to 13.85 ± 0.12 mL in 16 h (Figure 3), which is consistent with the literature [29]. At 16 h, the TTA of contaminated sourdough was significantly lower than that of the non-contaminated control sourdough (*p* < 0.05), which implies that the presence of *S. aureus* might inhibit the fermentation of sourdough.

### 3.4. The Effect of S. aureus Contamination on Metabolites

A total of 169 metabolites were identified, and the contents of 47 metabolites in fresh dough (Flo), non-contaminated control sourdough (Co16), and contaminated sourdough (S16) were significantly different (*p* < 0.05) (Appendix A). The metabolites with significantly different and >0.100 g/100 g are shown in Table 1.

The content of myo-inositol, phosphate, glycolic acid, fructose, sorbitol, glucose, xylose, lactose, γ-aminobutyric acid (GABA), succinic acid, and ribose significantly increased (*p* < 0.05) during fermentation. The content of 1-kestose, linoleic acid, oxoproline, sucrose, L-Malic acid, and galactinol significantly decreased (*p* < 0.05) during fermentation.

The myo-inositol are the hydrolysate of phytic acid, which can prevent absorption of minerals such as iron, potassium, magnesium, and zinc and in turn reduce their bioavailability. Flour contains phytases, and several LAB can excrete phytases [4]. The optimal pH of phytases is about 4.3, which is near the pH of sourdough and thus can be beneficial to hydrolysis of phytic acid [1]. The accumulation of glycolic acid during sourdough fermentation might be caused by the conversion of oxalate. Wholewheat flour is rich in oxalate, which can inhibit the absorption of minerals similarly to phytic acid [30]. The xylose and sorbitol are effective prebiotics [21]. Xylose was derived from pentosan hydrolysis; however, arabinose, another product of pentosan hydrolysis, had low content (<0.100 g/100 g, Appendix A). This might be due to the fact that arabinose can be easily utilized by LAB and/or *S. aureus*.

GABA was reported have several health benefits, such as reducing anxiety, reducing high blood pressure, and improving insomnia [31]. Accordingly, a large amount of literature focuses on developing GABA-enriched foods. Accumulation of GABA during wholewheat sourdough is a new finding of the present study. GABA content has been reported to be about 4.5 mg/100 g in dough and 13.67 ± 0.40 mg/100g in sourdough [18]. In this study, the GABA content in wholewheat fresh dough and sourdough were 24.2 ± 1.1 and 241 ± 17 mg/100 g, respectively. The content of GABA in wholewheat sourdough is higher than not only normal sourdough but also many typical GABA-enriched foods, such as germinated brown rice, which contained GABA 172 mg/100 g in one report [31]. GABA accumulation in wholewheat sourdough occurs partly because many LAB can synthesize GABA [18] and partly because grain bran contains high activity of glutamate decarboxylase (GAD) that can transfer glutamate into GABA [31].

In addition, fermentation reduced the content of atrazine-2-hydroxy (0.003 ± 0.005, 0.005 ± 0.001, and 0.018 ± 0.001 g/100g for S16, Co16, and Flo, respectively), which is the degradation product of atrazine, a common herbicide with a residue limit standard of 0.05 mg/kg (GB 2763-2019, China) [32]; however, there is no limit standard for atrazine-2-hydroxy. In any case, these results indicate that sourdough fermentation might improve the safety of the dough products.

Comparing the Co16 and S16, the content of metabolites that were significantly different (*p* < 0.05) included glucoheptose, sorbitol, succinic acid, glucoheptonic acid, sucrose, mannose, ribose, and kestose (Table 1 and Appendix A). After culturing, the content of mannose in *S. aureus*-contaminated sourdough decreased significantly (*p* < 0.05), but that in non-contaminated control sourdough increased significantly (*p* < 0.05). This might be because *S. aureus* could effectively utilize mannose [33,34] On the other hand, the sorbitol content showed the opposite trend, likely due to the fact that sorbitol is poorly transported by *S. aureus* [33]. Although neither sorbitol nor arabitol can be oxidized by *S. aureus*, these sugar alcohols can increase the rates of mannitol uptake [35]. The content of sorbitol in S16 was 2.2 and 37.4 times higher than that in Co16 and Flo, respectively. Thus, it is apparent that sorbitol is the characteristic metabolite in *S. aureus*-contaminated sourdough.

### 3.5. Analysis of VOMs

#### 3.5.1. Effect of *S. aureus* Contamination on VOMs in Sourdough

In total, 40 VOMs were identified, including 14 aldehydes, 11 alcohols, 7 acids, and 3 esters (Figure 4). They were found to be the main flavor compounds of sourdough [28]. The dominant aldehyde, acid, and alcohol were hexanal, butanoic acid, and 1-hexanol, respectively, which is consistent with the data reported in the literature [28,36]. The heptanoic acid, 3-methyl-1-butanol, 2-methyl-1-butanol, 3-methyl-hepta-1,6-dien-3-ol, and 2-octen-1-ol were detected only in the non-contaminated control sourdough.

Fresh dough contained the highest amounts of 3,4-dimethylpent-2-en-1-ol, tetrahydro-6-(2-pentenyl)-2H-pyran-2-one (also known as jasmine lactone), n-hexane, and butyl formate. However, their contents were significantly decreased during the fermentation process. The decrease in the contaminated sourdough was less, indicating that *S. aureus* could inhibit the fermentation of sourdough, which is consistent with the TTA (Figure 3). The content of n-hexane and 3,4-dimethylpent-2-en-1-ol in the non-contaminated control sourdough decreased to below the detectable limit after 4 h of fermentation. However, n-hexane could be detected throughout the fermentation process of contaminated sourdough.

Aldehydes provide sourdough with oil, fruit, and almond aromas. *S**. aureus* significantly reduced the content of hexanal, 2-hexenal, 2-ethylhexanal, 2-methylheptanal, and octanal but significantly increased the content of 2,4-decadienal, and 2-octenal (*p* < 0.05). The threshold of 2,4-decadienal was very low (0.007 ppb), and its off odor [37] might have a negative effect on the flavor of sourdough.

Alcohols, such as 1-hexanol, 1-pentanol, 1-octen-3-ol, 3-methyl-1-butanol, etc., are the main flavor compounds of sourdough that provide sourdough with fruit, wine, and fermented aroma [38]. *S. aureus* reduced the content of 2-methyl-1-butanol, 3-methyl-1-butanol, 2-octen-1-ol, 3-methyl-hepta-1,6-dien-3-ol, 1-hexanol, and 1-octen-3-ol while significantly increasing (*p* < 0.05) the content of 1-pentanol.

The content of acetic acid, butanoic acid, 2-methylbutanoic acid, hexanoic acid, 3-methyl-4-oxo-pentanoic acid, and heptanoic acid in the non-contaminated control sourdough were higher than those in contaminated sourdough, which is in agreement with the TTA (Figure 3). However, the content of 3-methylbutanoic acid in S16 was higher than that in Co16 (*p* < 0.05).

Esters provide sourdough with “fermented aroma”. Esters identified in sourdough were methyl formate, hexyl acetate, and butyl formate, and the presence of *S. aureus* reduced the content of these esters. Among these esters, hexyl acetate had a low content of 2 ppb, and it contributes to fruity and sweet flavors.

Furthermore, *S. aureus* significantly increased the content of furfural (*p* < 0.05); its threshold was higher than 3000 ppb and thus perhaps has no great influence on the flavor of sourdough. On the other hand, *S. aureus* decreased the content of 2-pentylfuran significantly (*p* < 0.05). The 2-pentylfuran can provide “fruit aroma” at a threshold of only 6 ppb and is the characteristic VOMs in sourdough [38]. Mandin et al. [39] reported that the 2-pentylfuran transferred from 2,4-decadienal, which is consistent with our results that the content of 2,4-decadienal in contaminated sourdough was higher (*p* < 0.05) than that in the non-contaminated control sourdough.

#### 3.5.2. PCA Analysis of VOMs

The PCA analysis of the data from Flo, Co4, Co8, S4, and S8 is shown in Figure 5. The first two principal components (PC1 and PC2) accounted for 71.5% of the total variance of the data. The contaminated sourdough and the non-contaminated control sourdough were obviously discriminated even at the culturing time of only 4 h. This indicates that the PCA model developed based on VOMs could be used as a rapid detection method for contamination of *S. aureus* in sourdough.

#### 3.5.3. Identification of Characteristic VOMs in Sourdough Contaminated by *S. aureus*

Comparing VOMs in S4 and Co4 showed that the content of n-hexane, 3,4-dimethylpent-2-en-1-ol, and butyl formate in S4 were higher (*p* < 0.05) than those in Co4. However, because these compounds are derived from flour, it appears that the characteristic VOMs produced by *S. aureus* were not found in S4.

Comparing VOMs in S8 and Co8 indicated that the content of 1-penten-3-ol, benzaldehyde, heptanal, 3,4-dimethylpent-2-en-1-ol, 2,3-dimethylundecane, furfural, and n-hexane in S8 were higher (*p* < 0.05) than those in Co8. Comparing VOMs in S16 and Co16 showed that the content of 1-pentanol, benzaldehyde, heptanal, 2,4-decadienal, 2,3-dimethylundecane, 2-hepten-1-ol, and 3-methylbutanoic acid in S16 were higher (*p* < 0.05) than those in Co16. Among those VOMs, n-hexane and 3,4-dimethylpent-2-en-1-ol were from flour, whereas benzaldehyde, heptanal, and furfural are the characteristic VOMs in sourdough. In addition, the content of 1-penten-3-ol in contaminated sourdough gradually decreased after 8 h, which was conversely correlated with the cell count of *S. aureus*. Thus, we might conclude that 1-pentanol, 2,3-dimethylundecane, and 3-methylbutanoic acid are the characteristic VOMs in *S. aureus*-contaminated sourdough.

The 3-methylbutanoic acid has also been reported to be the characteristic VOMs of *S. aureus* in various cultures and contaminated food matrixes. Tait et al. [40] identified that *S. aureus* specifically produces 3-methybutanoic acid in brain heart infusion broth, tryptic soy broth, and 1% glucose enteric fermentation-based broth. Chen et al. [41] described that 3-methylbutanoic acid and 3-methylbutanal are the volatile biomarkers for *S. aureus* in tryptic soy broth. Hu et al. [10] reported that 3-methy-butyric acid is the characteristic compound of *S. aureus* in the culture medium or in pork. A rapid detection method of *S. aureus* contamination in milk and pork based on 3-methylbutanoic acid content has also been reported [10,11]. 3-methylbutanoic acid could be detected when the population of *S. aureus* in pasteurized milk reached 10^6^ to 10^7^ cfu/mL, when that in raw milk reached 10^5^ to 10^6^ cfu/mL [11], and when that in pork reached 10^6^ to 10^7^ cfu/g [10]. In the present study, 3-methylbutanoic acid was also detected till the number of *S. aureus* cells reached 10^6^ cfu/g.

Similar to that of 3-methylbutanoic acid, the content of 1-pentanol and 2,3- dimethylundecane were positively correlated with the number of *S. aureus* cells in contaminated sourdough. Furthermore, the ratios of 1-pentanol and 2,3- dimethylundecane in the contaminated sourdough and the non-contaminated control sourdough were greater than that of 3-methylbutanoic acid. It appears that 1-pentanol and 2,3-dimethylundecane have higher sensitivity than 3-methylbutanoic acid, especially for 2,3-dimethylundecane, which was detected only in *S. aureus*-contaminated sourdough (S8 and S16). Despite all of the above, more studies are needed to confirm and to fully understand this finding.

## 4. Conclusions

The *S. aureus* could easily grow in sourdough to the level that exceeds the limit (10^3^ cfu/g) within 3 h, and the level of *S. aureus* contamination of only 10^2^ cfu/g can pose risks. The *S. aureus* contamination inhibited the sourdough fermentation. PCA model of volatile metabolites could be used to distinguish between contaminated sourdough and non-contaminated control sourdough that underwent culturing for 4 h. The sorbitol, 3-methylbutanoic acid, 1-pentanol, and 2,3-dimethylundecane might be used as the markers for sourdough that is contaminated with *S. aureus*.

## Figures and Tables

**Figure 1 foods-11-01960-f001:**
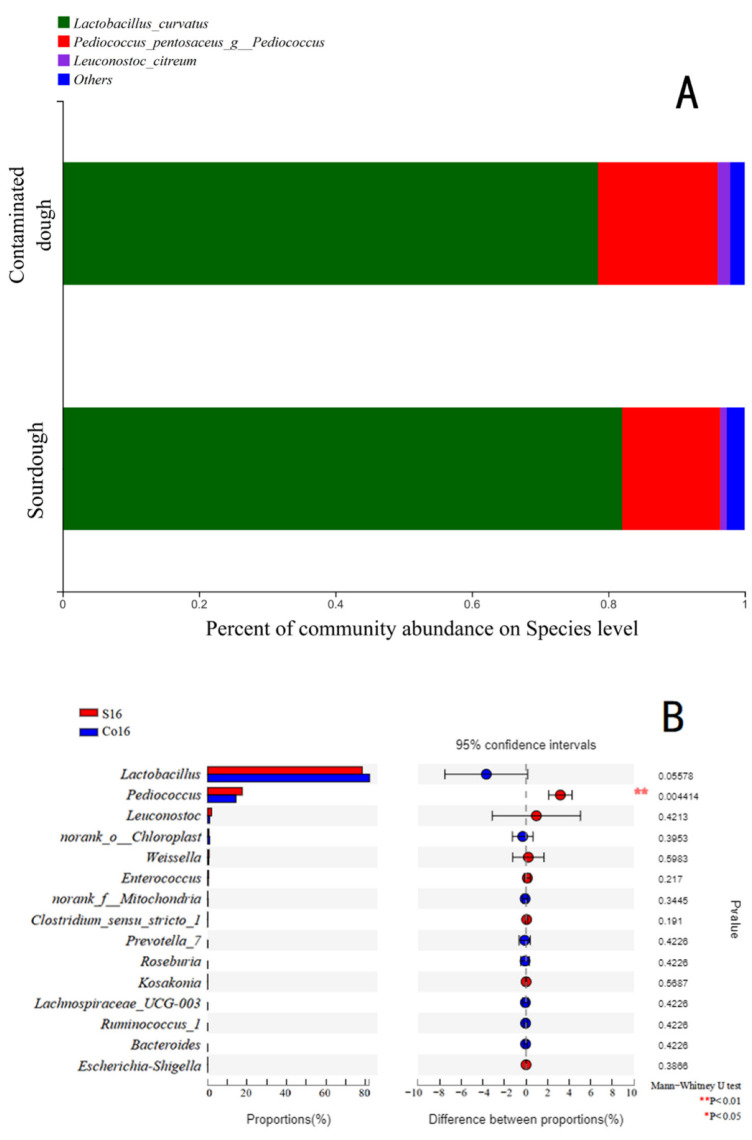
Bacterial composition of wholewheat sourdough (*n* = 6), with Co16 indicating wholewheat sourdough fermented for 16 h and S16 indicating wholewheat sourdough artificially contaminated with *S. aureus* and fermented for 16 h. ((**A**): Community barplot analysis; (**B**): Welch’s *t*-test bar plot on Genus level). Co16 indicating wholewheat sourdough fermented for 16 h. S16 indicating wholewheat sourdough artificial contaminated with *S. aureus* and cultured for 16h.

**Figure 2 foods-11-01960-f002:**
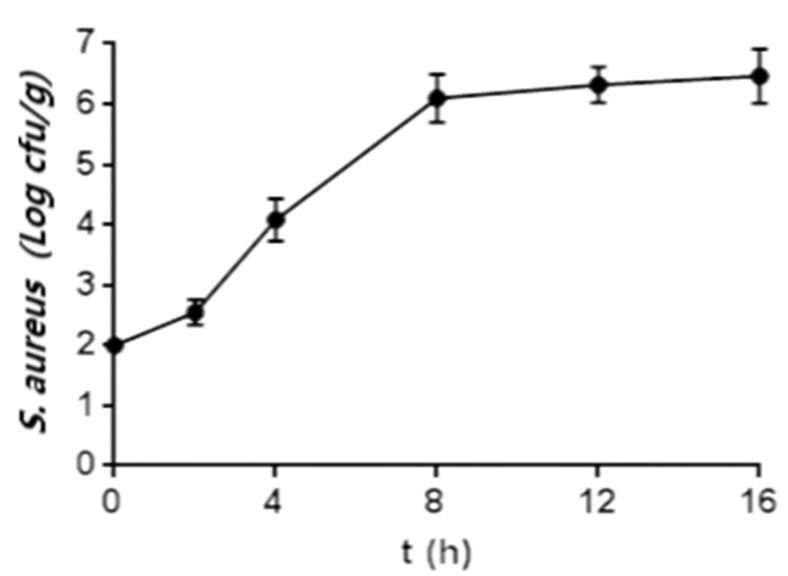
Growth curve of *S. aureus* in fermenting wholewheat sourdough (*n* = 6).

**Figure 3 foods-11-01960-f003:**
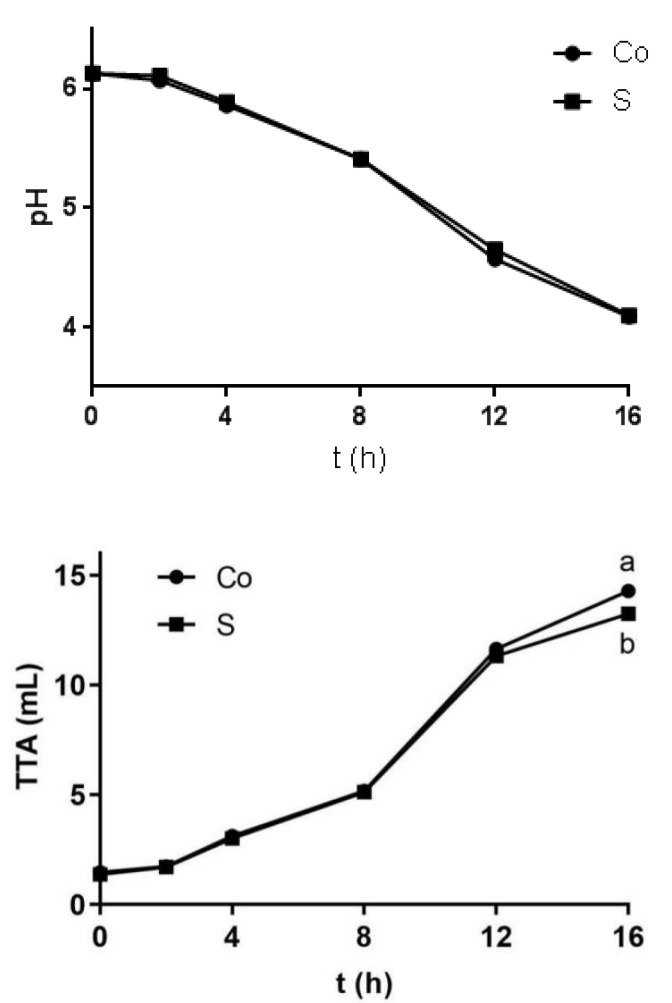
The effect of *S. aureus* contamination on pH and TTA of sourdough, with Co and S indicating wholewheat sourdough and wholewheat sourdough artificial contaminated with *S. aureus* (10^2^ cfu/g); ^a,b^ means in each bar having different letters are significantly different (*p* < 0.05, *n* = 6).

**Figure 4 foods-11-01960-f004:**
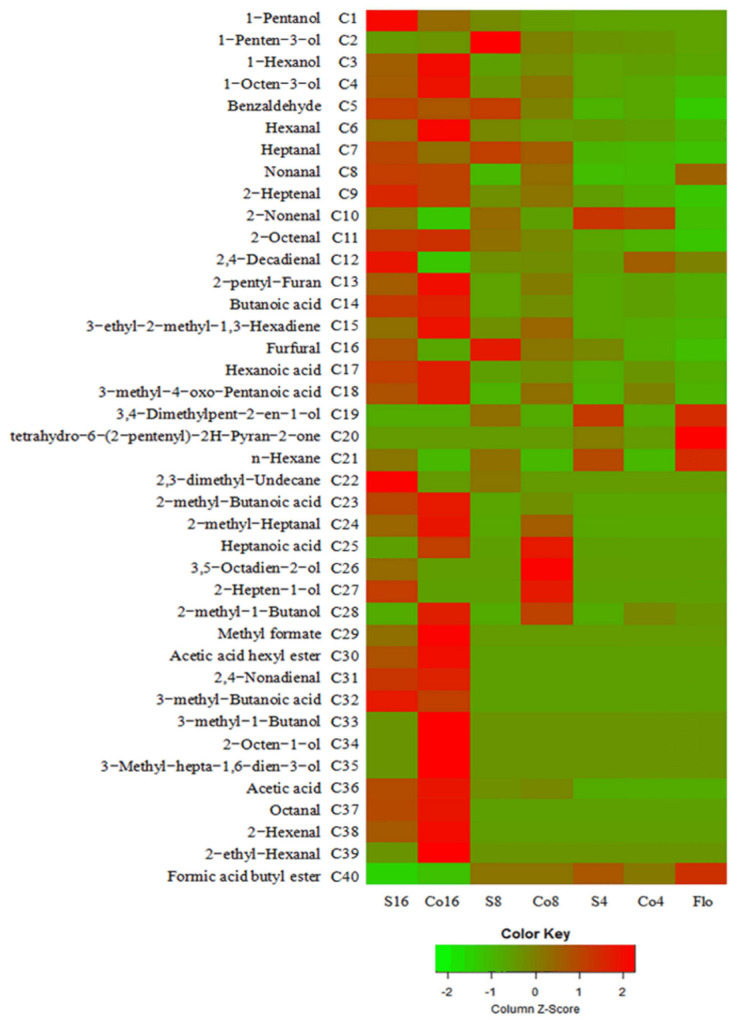
Heatmap of VOC in sourdough. Flo indicating fresh dough. Co4, Co8 and Co16 indicating wholewheat sourdough fermented for 4, 8 and 16 h, respectively. S4, S8 and S16 indicating wholewheat sourdough artificial contaminated with *S. aureus* (10^2^ cfu/g) and cultured for 4, 8 and 16 h, respectively (*n* = 6).

**Figure 5 foods-11-01960-f005:**
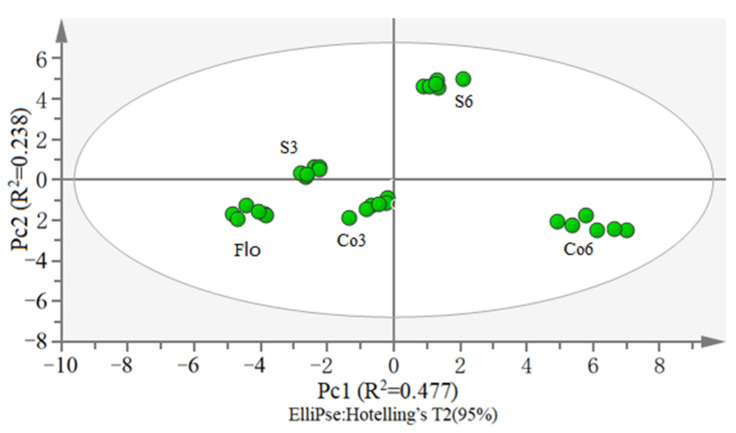
PCA analysis of VOC in sourdough. Flo indicates fresh dough without fermentation. Co4 and Co8 indicate wholewheat sourdough fermented for 4 and 8 h, respectively. S4 and S8 indicate wholewheat sourdough artificially contaminated with *S. aureus* (10^2^ cfu/g) and cultured for 4 and 8 h, respectively (*n* = 6).

**Table 1 foods-11-01960-t001:** Key metabolites (g/100 g) of fresh dough, sourdough, and sourdough artificially contaminated with *S. aureus*.

Metabolites	Flo	Co16	S16
Fructose	1.041 ± 0.022 ^b^	5.225 ± 0.432 ^a^	5.488 ± 1.339 ^a^
Glycolic acid	0.008 ± 0.001 ^b^	3.989 ± 0.313 ^a^	4.870 ± 1.370 ^a^
Myo-inositol	0.166 ± 0.011 ^b^	1.741 ± 0.151 ^a^	2.044 ± 0.356 ^a^
Phosphate	0.122 ± 0.006 ^b^	1.126 ± 0.079 ^a^	1.173 ± 0.148 ^a^
Sorbitol	0.027 ± 0.002 ^c^	0.463 ± 0.038 ^b^	1.010 ± 0.231 ^a^
Glucose	0.033 ± 0.000 ^b^	0.649 ± 0.042 ^a^	0.757 ± 0.182 ^a^
Xylose	0.014 ± 0.001 ^b^	0.659 ± 0.056 ^a^	0.751 ± 0.128 ^a^
Lactose	0.009 ± 0.001 ^b^	0.326 ± 0.065 ^a^	0.345 ± 0.124 ^a^
Kestose	0.933 ± 0.166 ^a^	0.422 ± 0.034 ^b^	0.353 ± 0.017 ^c^
γ-Aminobutyric acid (GABA)	0.024 ± 0.001 ^b^	0.241 ± 0.017 ^a^	0.235 ± 0.016 ^a^
Succinic acid	0.028 ± 0.001 ^c^	0.144 ± 0.014 ^b^	0.186 ± 0.020 ^a^
Linoleic acid	0.234 ± 0.006 ^a^	0.112 ± 0.011 ^b^	0.131 ± 0.006 ^b^
Ribose	0.002 ± 0.001 ^c^	0.106 ± 0.002 ^a^	0.099 ± 0.003 ^b^
Mannose	0.134 ± 0.003 ^b^	0.278 ± 0.086 ^a^	0.057 ± 0.014 ^c^
Oxoproline	0.114 ± 0.005 ^a^	0.065 ± 0.002 ^b^	0.064 ± 0.003 ^b^
L-Malic acid	1.154 ± 0.034 ^a^	0.003 ± 0.001 ^b^	0.001 ± 0.001 ^b^
Sucrose	0.218 ± 0.015 ^a^	0.054 ± 0.004 ^b^	0.037 ± 0.005 ^c^
Galactinol	0.110 ± 0.008 ^a^	0.032 ± 0.002 ^b^	0.034 ± 0.005 ^b^
Atrazine-2-hydroxy	0.018 ± 0.001 ^a^	0.005 ± 0.001 ^b^	0.003 ± 0.005 ^b^

^a–c^ Means in each row having different letters are significantly different (*p* < 0.05, *n* = 6). Flo, Co16, and S16 indicate fresh dough, sourdough fermented for 16 h, and sourdough artificially contaminated with *S. aureus* (10^2^ cfu/g) cultured for 16 h, respectively.

## Data Availability

The data presented in this study are available on request from the corresponding author.

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
