# Peer review of "Effect of Staphylococcus aureus Contamination on the Microbial Diversity and Metabolites in Wholewheat Sourdough"

_foods, 2022, doi:10.3390/foods11131960_

Round 1

Reviewer 1 Report

Comments to Authors,

The manuscript entitled "Effect of fermentation and Staphylococcus aureus CICC21600 contamination on the microbial diversity and metabolites in wholewheat sourdough" is suitable for publication in Foods; however, it has to be improved in some aspects, considering the following general and specific comments.

General comments:

The authors mention in the title the effect of Staphylococcus aureus CICC21600 contamination on the microbial diversity and metabolites of whole wheat sourdough; however, in the Introduction section, they only mention an overview of the Staphylococcus aureus bacteria, and do not mention anything about this strain (Staphylococcus aureus CICC21600). What is the importance then of this strain (Staphylococcus aureus CICC21600) in this type of food products? And if so, then the authors should mention more details in the Introduction section so that the use (effect) of this strain of Staphylococcus aureus is justified.

Specific Comments:

Lines 13 and 14: These two sentences seem unconnected. Authors should combine both sentences into one and take care of the syntax.

Line 186: It should be read as, … and > 0.100 g/100 g were…

Line 224: It should be read as, 3.4.1. Effect of fermentation and Staphylococcus aureus CICC21600 contamination on VOMs in sourdough

Reviewer 2 Report

The manuscript contains interesting information on the microbiological challenge test of wholewheat sourdough using Staphylococcus aureus as the test organism. The microbial community of the sourdough was studied. Also, the metabolites of the contaminated sourdough was compared with the control. However, the manuscript requires a re-work to make it of better quality worth publishing.

Some suggestions that could improve the work have been highlighted and noted within the manuscript. 

Round 2

Reviewer 2 Report

I commend the effort of authors to improve the manuscript.

Author Response

We sincerely thank our reviewers for their suggestions